# The Efficacy and Utility of Lower-Dimensional Riemannian Geometry for EEG-Based Emotion Classification

Zubaidah Al-Mashhadani [1], Nasrin Bayat [1], Ibrahim F. Kadhim [2], Renoa Choudhury [2] and Joon-Hyuk Park [2,*]

1   Department of Electrical and Computer Engineering, University of Central Florida, 4000 Central Florida Blvd, Orlando, FL 32816, USA; zubaidah@knights.ucf.edu (Z.A.-M.); nasrinbayat@knights.ucf.edu (N.B.)
2   Department of Mechanical and Aerospace Engineering, University of Central Florida, 4000 Central Florida Blvd, Orlando, FL 32816, USA; ibrahimkadhim@knights.ucf.edu (I.F.K.); renoa@knights.ucf.edu (R.C.)
*   Correspondence: joonpark@ucf.edu; Tel.: +1-407-823-1227

**Abstract:** Electroencephalography (EEG) signals have diverse applications in brain-computer interfaces (BCIs), neurological condition diagnoses, and emotion recognition across healthcare, education, and entertainment domains. This paper presents a robust method that leverages Riemannian geometry to enhance the accuracy of EEG-based emotion classification. The proposed approach involves adaptive feature extraction using principal component analysis (PCA) in the Euclidean space to capture relevant signal characteristics and improve classification performance. Covariance matrices are derived from the extracted features and projected onto the Riemannian manifold. Emotion classification is performed using the minimum distance to Riemannian mean (MDRM) classifier. The effectiveness of the method was evaluated through experiments on four datasets, DEAP, DREAMER, MAHNOB, and SEED, demonstrating its generalizability and consistent accuracy improvement across different scenarios. The classification accuracy and robustness were compared with several state-of-the-art classification methods, which supports the validity and efficacy of using Riemannian geometry for enhancing the accuracy of EEG-based emotion classification.

**Keywords:** brain–computer interface; Riemannian geometry; emotion recognition; PCA; electroencephalogram (EEG); feature extraction; dimensionality reduction; Fréchet mean

## 1. Introduction

Emotions are complex psychophysiological processes that encompass cognitive, physiological, and behavioral responses to internal and external stimuli. They can be expressed through various channels, including language, facial expressions, tone of voice, and gestures. Emotions are influenced by factors such as motivation, personality, temperament, physical conditions, and mood. Understanding emotions is essential for gaining insights into human behavior and enhancing communication, well-being, and emotional intelligence [1–3]. The origins of emotions lie in neural activities within specific brain regions, where varying levels of neural stimulation can evoke a range of emotional responses [4–6]. Electroencephalography (EEG) signals provide a means to monitor the electrical activity in the central nervous system, allowing us to observe the fluctuations, locations, and functional interconnections of brain impulses. By analyzing these signals, we gain direct access to an individual's inner state of mind, aiding in the understanding of human behavior and decision-making processes. This valuable insight enables the development of computational models and systems that can effectively comprehend and respond to human emotions [7].

Emotion classification from EEG signals often involves using spatial filtering and supervised classification algorithms [8]. Spatial filters, such as those employed in the common spatial pattern (CSP) algorithm, use a linear combination of the EEG signals to maximize

the discriminability of two classes [9]. For example, the variance of the spatially filtered EEG signals can be used as features for classification through linear discriminant analysis (LDA) [10,11]. On the other hand, tangent space mapping creates a higher dimensional space that allows the acquisition of more spatial information [12]. However, this comes at a high computational cost, limiting the implementation of some algorithms that could lead to bias or overfitting [10,13,14]. Such spatial filtering techniques used on EEG signals rely heavily on sophisticated computations based on covariance matrices estimated from EEG signals, which can be employed directly as the features of interest in the Riemannian framework. Riemannian manifold has unique advantages:

(i) Direct computations on the covariance matrices of EEG signals through applying differential geometry tools and algorithms, such as minimum distance to Riemannian mean (MDRM). Thus, it facilitates the classification of covariance matrices within the Riemannian space to help enhance EEG signal analysis [10].

(ii) Riemannian geodesic distance metric accounts for the geometry of the space of covariance matrices. It is invariant by projection, allowing the use of dimensionality reduction techniques, such as principal component analysis (PCA), to compute the space of covariance matrices without losing essential information or distorting the structure of the space, which is crucial for accurate analysis of EEG signals [10,15,16].

(iii) The dimensionality reduction performed on Riemannian spaces offers a means to exploit high dimensional and more discriminative features, subsequently improving accuracy in classification or clustering [17].

(iv) The Riemannian framework can handle intra-individual variability by modeling individual-specific covariance matrices, which can capture variations in brain activity patterns over time. Similarly, it can address inter-individual variability by employing population-based covariance models that capture commonalities across individuals.

(v) The Riemannian framework is robust to changes in electrode placement and noise handling, allowing for reliable and accurate analysis of EEG signals [16].

(vi) The Riemannian framework is insensitive to spatial filtering of the data, resulting in improved classification accuracy.

In addition, applying Euclidean geometry to the space of symmetric and positive definite (SPD) matrices yields an artifact known as the swelling effect [16,18,19]. The Riemannian framework can prevent this by accounting for the unique geometry of the space of covariance matrices while processing these matrices in their native space, allowing a direct implementation of the Fréchet mean algorithm [10]. Moreover, the Riemannian structure induced by the affine-invariant Riemannian metric (AIRM), which ensures that the distance between these different representations of covariance matrices is equivalent [20], overcomes the constraints of Euclidean geometry in exchange for increased computation cost as the size of the manifold increases [17].

Despite the recent advancements in emotion recognition using facial expressions, speech, and EEG signals, our understanding of human emotions remains insufficient. This highlights the need for new innovations in human–computer interfaces (HCIs) to gain deeper insights into human emotional responses to specific stimuli.

Emotion recognition using Riemannian metrics currently lacks comprehensive comparative studies across multiple datasets as a benchmark. Moreover, the existing work in emotion classification predominantly focuses on a single dataset, which limits its translational applicability. Additionally, the challenge of handling intra-subject variability, where the features of interest vary between subjects and instances, has not been adequately addressed. To address these limitations, this study aims to investigate and compare the performance of classification methods within the Riemannian framework, traditional feature selection, and a novel dynamic feature selection approach across four diverse datasets. The primary objective is to validate the robustness and generalizability of the proposed method, enabling its potential application in real-world scenarios.

Our work aims to develop a dynamic feature selection framework within the minimum distance to Riemannian mean (DFS-MDRM) approach for classifying human emotions

and investigating the impact of feature selection on model performance. The framework involves two main steps. In the first step, we preprocess EEG signals by applying filtering and normalization techniques, followed by the utilization of shrinkage covariance matrix estimation to transform the extracted features onto the Riemannian manifold. In the second step, we employ the SPD matrices on the manifold to perform supervised classification using the MDRM method. This entails estimating the mean of each class using the Fréchet mean and computing distances on the manifold using AIRM. To evaluate the effectiveness of our approach, we conducted four experiments across five frequency bands. Validation was performed using leave-one-out cross-validation (LOOCV), where we employed dynamic principal component analysis (PCA) to identify the most appropriate features for classifying each test trial, ensuring adaptability across different datasets and mitigating the impact of variability in observations. The approach presented in this work leverages the advantages of the Riemannian framework to enhance classification accuracy. It better accounts for the specific geometry of the space of covariance matrices reduces the impact of irrelevant features, and optimizes the model parameters.

In this work, we make two main contributions:

(1) Integration of traditional feature extraction techniques, specifically principal component analysis (PCA), into a dynamic feature extraction process. By representing the extracted features as covariance matrices and leveraging their distinctive characteristics in the Riemannian manifold space, our proposed method effectively addresses variabilities observed across different instances.
(2) Demonstration of the generalizability and robustness of the proposed method through a successful application to four well-known datasets with varying characteristics. The achieved results outperformed state-of-the-art methods, highlighting the considerable potential of this approach for practical applications.

The rest of the paper is organized as follows: Section 2 contains the relevant work in emotion classification. In Section 3, we introduce the basic concepts of Riemannian geometry, discuss the primary algorithms, and present our complete methodology. Section 4 reports the results of subject-dependent EEG signal analysis. In Section 5, we provide discussions and outline future work. Finally, in Section 6, we conclude and present perspectives on our work.

## 2. Related Work

Emotion classification using EEG signals involves several stages: emotional induction, signal acquisition, signal preprocessing, feature extraction, dimensionality reduction, emotion recognition, and classification. Emotional induction methods often rely on visual and auditory stimuli like emotional videos [21], music videos [1], or film clips [3,22].

Feature extraction plays a crucial role in identifying emotions from EEG signals [23]. Prior studies have demonstrated a strong correlation between EEG emotions and EEG frequencies [24,25]. Typically, EEG frequency features are extracted by mapping the EEG signals to Theta (4–7 Hz), Alpha (8–13 Hz), Beta (14–29 Hz), Gamma (30–47 Hz), and other frequency bands [26]. Dimensionality reduction and classification are also essential stages in the emotion classification pipeline [27]. Commonly used dimensionality reduction techniques include linear discriminant analysis (LDA) [28] and principal component analysis (PCA) [29].The key to accurate emotion recognition from EEG signals lies in extracting emotion-related features and optimizing the performance of the classifier.

Deep learning and machine learning state-of-art techniques have been widely used for this purpose. Pereira et al. [30] conducted emotion classification on the DEAP dataset using Support Vector Machine (SVM) classifiers with both higher-order crossing (HOC) and power spectral density (PSD) features. The leave-one-subject-out classification approach achieved average accuracies of 53.42% for valence and 52.03% for arousal. Kraljević et al. [31] used a support vector machine with the Gaussian kernel evaluated on the DEAP dataset. The accuracy scores obtained after using PCA achieved 60.59% for valence and 62.45% for arousal. Jirayucharoensak et al. [28] applied PCA for feature extraction and implemented

covariance shift adaptation of the principal components, using a deep learning network (DLN) as the classifier, which resulted in accuracies of 49.52% and 46.03% for valence and arousal, respectively. Similarly, Ben Henia et al. [32] also adopted SVM for MAHNOB dataset emotion classification, obtaining 59.75% and 57.44% for arousal and valence, respectively.

Muhammad et al. [33] developed a multimodal emotion recognition method combining EEG and facial video clips. Using DCCA, they fused features extracted by ResNet50 (for facial video) and 1D-CNN (for EEG). Classification of happy, neutral, and sad emotions was performed using the SoftMax classifier. The method was validated on MAHNOB HCI and DEAP datasets. Song et al. [34] developed a model that integrates efficient channel attention into a combination of CNN and gated circulation units for emotion recognition on the DEAP dataset. Wang et al. [35] proposed a CNN model for cross-subject emotion recognition and selecting the channels with the highest accuracy as the feature vector, the model is validated on the SEED dataset.

Riemannian geometry has been exploited for emotion recognition [36]. Proposed the utilization of the Riemannian manifold and PCA for speech emotion recognition. Liu et al. [37] implemented a visual-based emotion recognition model using the Riemannian manifold. Abdel-Ghaffar et al. [7] developed a subject-dependent binary emotion classification of valence and arousal using transfer learning MDRM framework, and validated the model on the DEAP dataset. Kim et al. proposed a Riemannian-based deep learning network that generates discriminative features for EEG classification on the DEAP dataset, achieving an accuracy of 55.2% [38]. Zhang et al. [39] proposed deep neural architecture, utilizing a deep long short-term memory (LSTM) network with a soft attention mechanism to learn temporal information from EEG signals and validated it on the SEED dataset.

In this work, we introduce and validate an EEG-based emotion classification framework that utilizes Riemannian geometry, which integrates data normalization, dimensionality reduction, and hyperparameter tuning techniques. The objective of the proposed approach is to demonstrate the efficacy and utility of Riemannian geometry applied to classification problems. The classification accuracy of the proposed approach was validated on multiple high-dimensional datasets and we compared its performance with state-of-the-art machine learning models.

## 3. Methods

This section provides a brief introduction to the primary techniques and algorithms utilized in this study, as well as an explanation of each proposed method.

### 3.1. Riemannian

In the Riemannian framework, the distances are geodesics, the shortest distance in the form of a curve on the surface of a manifold between two points, as shown in Figure 1. In the context of EEG signals, the variable $X$ will be referred to as an EEG signal in $\mathcal{R}^{C \times N}$ where $C$ denotes the number of channels, the electrodes placed on the subject's cap, and $N$ denotes the number of samples (observations) recorded by the system. The covariance matrix is estimated by Equation (1) [16].

$$\Sigma = \frac{1}{N-1} X X^T \qquad (1)$$

Covariance matrices defined on a manifold are SPD matrices in $\mathcal{R}^{C \times C}$ that satisfy symmetry; all eigenvalues are non-negative, given $\Sigma = \Sigma^T$ [10]. The mean of the EEG signals represented by covariance matrices is computed using Frechét mean, a method for calculating the mean of data on a manifold [10]. It is also known as the geometric or Riemannian mean and is calculated using the geodesic distance on a manifold. The Fréchet mean is used to reduce the influence of outliers and provides a more representative estimate of the central tendency of the data on the manifold [40]. Considering the trials of

an experiment of a specific class, as a set of covariance matrices denoted as $\{\Sigma_i\}_i = 1\ldots I$, the mean or center of the class is expressed as follows:

$$\bar{\Sigma} = arg \min_{\Sigma \in \mu} \sum_{i=1}^{I} d(\Sigma_i, \Sigma)^2. \tag{2}$$

When estimating the mean of SPD matrices, the choice of the distance metric is crucial to the resultant mean. In this work, the AIRM is used to measure the distance between covariance matrices, and is then used to compute the Fréchet mean.

$$d_{AIR} = \|Log(\Sigma_1^{-1/2}\Sigma_2\Sigma_1^{-1/2})\|_F \tag{3}$$

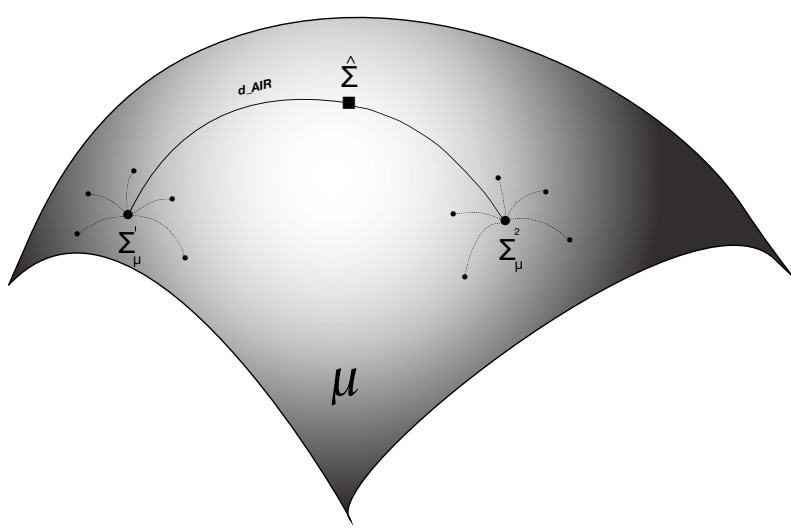

**Figure 1.** MDRM principle for a binary problem on a Riemannian manifold.

### 3.2. Classification Algorithms

The emotion classification approach is based on the minimum distance to Riemannian mean (MDRM), as illustrated in Figure 1. The MDRM principle is applied to a binary classification problem in a curved space, where the mean of each class covariance matrix represents the class center or the center of mass [7,16,41]. The MDRM classification using the Fréchet mean and AIRM is employed to classify a test trial, as described in Algorithms 1 and 2. The test trial is assigned to the class mean with the shortest distance to it in the Riemannian space. This approach improves classification accuracy by using a distance metric that accounts for the specific geometry of the space of covariance matrices, as implemented in Algorithm 2.

The Fréchet mean of class $k$ is computed for each class in Algorithm 1, then MDRM is used for predicting the unknown class for the test trial. The test trial is classified to the class mean with the shortest distance, in this case, $\Sigma_{\mu}^{(1)}$, as shown in Figure 1.

---

**Algorithm 1** Estimation of Riemannian centers of classes.

---

**Input:** a set of labeled trials $\mathbf{X}_i$ for $i = 1, ..., I$.
**Input:** $I^{(k)}$, a set of indices of trials of class $k$.
**Output:** $\Sigma_{\mu}^{(k)}$, $k = 1,\ldots,K$, centers of classes.
  1: Compute covariance matrices $\hat{\Sigma}_i$ of $X_i$.
  2: **for** $i = 1 \to K$ **do**
  3:     $\Sigma_{\mu}^{(k)} = \mu\,(\hat{\Sigma}_i, i \in I^{(k)})$
  4: **end for**
  5: **return** $\Sigma_{\mu}^{(k)}$

---

---

**Algorithm 2** Minimum distance to Riemannian mean.

---

**Input:** a set of $\mathbf{X}_i$ of $K$ different known classes.

**Input:** $\mathbf{X}$ an EEG trial of unknown class.

**Input:** $\Sigma_\mu^{(k)}$, K centers of classes from Algorithm 1.

**Output:** $\hat{k}$ the predicted class of test trial $\mathbf{X}$

  1: Compute covariance matrix $\hat{\Sigma}$ of $X$.

  2: **for** $i = 1 \rightarrow K$ **do**

  3:     $\hat{k} = arg\ min_k \delta(\hat{\Sigma}, \Sigma_\mu^{(k)})$

  4: **end for**

  5: **return** $\hat{k}$

---

The unique structure of covariance matrices belongs to the Riemannian manifold of the symmetric positive definite, allowing manipulation within the Riemannian geometry framework. Hence, the manifold of PSD matrices allows for the use of explicit formulas for essential operations in the Riemannian manifold to easily implement the algorithms.

### 3.3. Methods

This study uses three primary methods to investigate the impact of using solely Riemannian geometry for classification and the effect of employing dimensionality reduction and hyperparameter tuning on the classification results. The details of these methods are presented in this section. A block diagram of the proposed emotion classification method using Riemannian geometry is shown in Figure 2, illustrating the key steps involved in the classification process. These steps include preprocessing EEG data, dimensionality reduction, feature extraction, and classification using the Riemannian manifold.

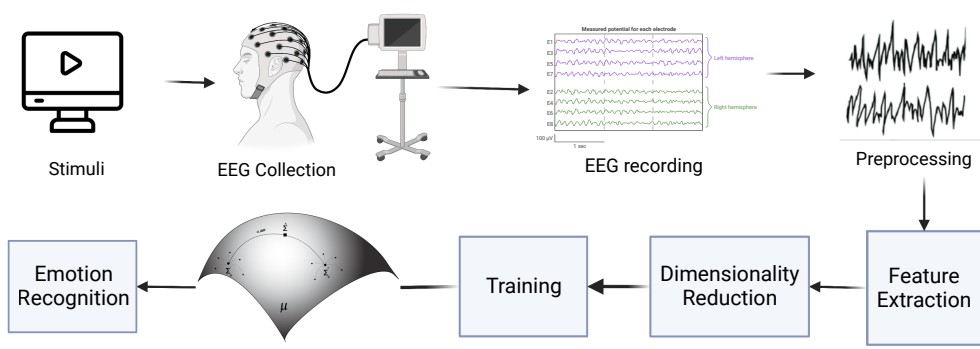

**Figure 2.** Block diagram of proposed EEG-based emotion classification method using Riemannian geometry.

### 3.3.1. Method 1 (MDRM)

This method provides the fundamental framework for the subsequent approaches. First, the data undergo preprocessing, which includes filtering, normalization, and transformation into covariance matrices. Subsequently, the MDRM algorithm is applied for classification purposes, whereby the mean of each class is determined using the Fréchet mean, and the distance is measured using the AIRM. The performance evaluation is conducted using the leave-one-out cross-validation technique to determine the average accuracy.

### 3.3.2. Method 2 (MDRM plus PCA)

Method 2 extends the approach used in method 1 by incorporating dimensionality reduction through PCA. This stage includes feature selection to improve accuracy by eliminating irrelevant features, reducing noise, and mitigating overfitting, resulting in a more robust and generalizable model.

### 3.3.3. Method 3 (MDRM plus PCA plus Hyperparameter Tuning)

Method 3 builds upon the foundation of method 2 as mentioned earlier. Specifically, method 3 incorporates hyperparameter tuning to optimize the performance of the classification model. The approach evaluates the model with 20 hyperparameter combinations, consisting of 5 frequency bandwidths (theta, alpha, beta, gamma, and all) and four threshold values for principal components. The proposed method assigns each test trial a unique set of hyperparameters based on the dataset's characteristics during the validation process. This approach is illustrated in Algorithm 3 and is expected to exhibit superior classification accuracy compared to method 1 and 2. In addition to utilizing the MDRM algorithm and PCA, hyperparameter tuning was introduced in this study. Selecting the frequency band and threshold to determine the components required to explain a certain percentage of the total variation in the PCA process results in the number of feature vectors used for classification, which depends on the threshold chosen for each test trial.

The hyperparameter selection process is achieved using inner and outer leave-one-out cross-validations (LOOCV). The inner cross-validation loop evaluates the hyperparameter combinations by taking a validation trial and measuring each hyperparameter combination's classification accuracy. The highest accuracy of the hyperparameter combination is applied for each test trial within the outer cross-validation loop. The procedure for selecting hyperparameter combinations involved identifying the frequency band with the highest accuracy across all thresholds, followed by selecting the highest threshold within that band within the inner cross-validation to ensure no overfitting occurs. For each test trial, feature vectors were pre-processed using the optimal hyperparameters, and the covariance matrix form for each trial was computed. The mean of each class was then calculated using the Riemannian mean algorithm, as shown in Algorithm 1 [42].

---

**Algorithm 3** Feature extraction and hyperparameter tuning.

---

**Input:** $k1$ = outer folds, $k2$ = inner folds.
**Input:** $D$, the subject dataset.
**Input:** $P_i^{pair}$, hyperparameters pairs.
**Output:** Accuracy
  1: **for** $i = 1 \rightarrow k1$ **do**
  2:      Split $D$ into $D_i^{train}$, $D_i^{test}$ for the $i$-th split
  3:      **for** $p \in P_i^{pair}$ **do**
  4:          **for** $j = 1 \rightarrow k2$ **do**
  5:              Split $D_i^{train}$ into $D_j^{train}$, $D_j^{val}$.
  6:              Apply BPF $(\theta, \alpha, \beta, \gamma)$.
  7:              Apply PCA.
  8:              Compute $\hat{\Sigma}_i$ of $X_i$.
  9:              Compute $\Sigma_\mu^{(k)}$ in Algorithm 1.
10:              Predict $\hat{k}$ in Algorithm 2.
11:              **Return** $\hat{k}$.
12:          **end for**
13:          Compute the accuracy with $D_j^{test}$.
14:      **end for**
15:      Select the optimal hyperparameter combination and apply it to $D_i^{test}$.
16:      Predict $\hat{k}$ using $D_i^{test}$ in Algorithm 2.
17:      % $\hat{k}$ is the predicted class of the test trial.
18: **end for**
19: **return** *Accuracy*

---

### 3.4. Pre-Processing and Feature Extraction

The utilized data preprocessing techniques are normalization and filtering. First, temporal filtering using a Butterworth bandpass filter was applied to extract frequency

bands of interest. Next, the data was normalized using the Frobenius norm, as defined in Equations (4) and (5), accounting for variations in amplitude across trials allowing to calculate the total energy or overall strength of the signal separately for each subject, ensuring that the analysis was not biased towards any particular trial or individual.

The variables used in the equations were also defined: $CH$ denotes the number of channels that represent the feature vectors, $L$ represents the length of each trial, and $T$ represents the number of trials. $A_i$ represents the Frobenius norm of the $i$-th feature vector, $n$ represents the number of electrodes, and $X$ represents the input data.

$$\eta = \sqrt{\frac{\sum_{i=1}^{n} \|A_i\|_F}{T \times L \times CH}} \qquad (4)$$

$$Normalized\ Data = \frac{X}{\eta} \qquad (5)$$

Feature extraction is performed using PCA, a statistical technique that identifies essential features in a dataset and represents them as new variables called principal components [43,44]. These components are selected to capture as much variation in the data as possible, and their weights are determined by calculating the eigenvectors and eigenvalues of the data covariance matrix. By selecting only the principal components with the highest eigenvalues, the dimensionalities of the data are reduced while retaining as much information as possible [45].

*3.5. Datasets*

This section contains a brief description of the datasets that were used and tested in this work. The study focuses on the classification of two targeted emotions, namely valence, and arousal, using EEG signals. Valence ranges from unpleasant (unhappy) to pleasant (happy), whereas arousal ranges from inactive (calm) to active (excited), as shown in Figure 3 [1]. The self-assessment was carried out through the self-assessment manikins (SAM), a well-known visualization for the scales numbered 1–9 for emotion rating as shown in Figure 4 [46]. A threshold was applied uniformly within each dataset to separate labels into high/low categories for binary classification. During the experiments, the electrodes were placed according to the international system 10–20 system.

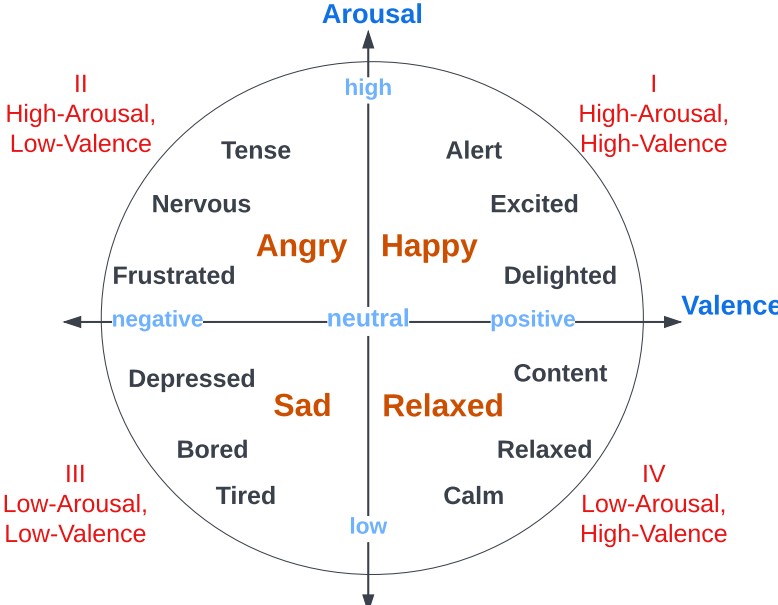

**Figure 3.** Valence arousal circumplex for emotion classification.

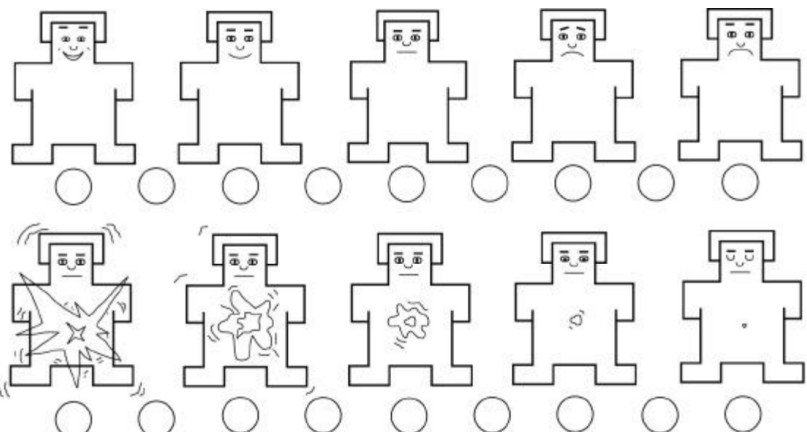

**Figure 4.** SAM, the self-assessment manikin for valence and arousal [46].

### 3.5.1. DEAP Dataset

The DEAP dataset is a multi-modal emotional dataset; the emotions elicited are valence, arousal, and dominance for 32 participants. Emotions are elicited using the response to music video clip stimuli; the EEG was recorded at a sampling rate of 512 Hz using 32 electrodes. The experiment recordings include electrocardiogram (ECG), galvanic skin response (GSR), and electromyogram (EMG) [1]. A one-minute segment of 40 music videos was presented to the subject through a web-based emotion assessment interface. The participants watched the music videos and rated them on a nine-point valence, arousal, and dominance scale.

### 3.5.2. DREAMER Dataset

The emotions are valence, arousal, and dominance, elicited by audio-visual stimuli in the form of film clips. The session recordings contain EEG and electrocardiogram (ECG) data from 23 participants while watching 18 film clips. The EEG was recorded at a 128 Hz sampling rate using 16 electrodes [22].

### 3.5.3. MAHNOB Dataset

In the MAHNOB database, 32 EEG channels were used to record a multi-modal dataset for 27 subjects at 256 Hz. The experiment entailed watching 20 films and evaluating the valence, arousal, and dominance of the felt emotions on a scale of 1 to 9 after each film clip. The experiment data also comprised simultaneous eye gaze recordings, physiological signals, audio and verbal expressions, and videos of facial recordings [21].

### 3.5.4. SEED Dataset

The EEG dataset was collected from 15 participants and targeted three positive, neutral, and negative emotions. The adapted stimuli during the experiment were emotional movie clips, where each emotion had five corresponding emotional clips. Fifteen films were presented to the subjects to elicit their emotions; each film was approximately 4 min long [3].

## 4. Results

This section thoroughly evaluates the effectiveness of classifying the emotion EEG signals using Riemannian geometry. It demonstrates the impact of dimensionality reduction and hyperparameter tuning techniques on four datasets. Furthermore, a comparison of the performance of the proposed method with that of existing state-of-the-art models is presented, including the K-nearest neighbors (KNN) algorithm and convolutional neural network (CNN).

### 4.1. DEAP Dataset

A comprehensive comparison of the proposed method's classification performance with KNN and CNN models in terms of average accuracy for the valence and arousal classes on the DEAP dataset is presented in Table 1. The results demonstrate that the accuracy of the proposed method 1 outperforms CNN, while method 2 achieves higher accuracy by integrating dimensionality reduction and outperforms both KNN and method 1. However, method 3, which employs both dimensionality reduction and hyperparameters tuning, achieves the highest performance, with an accuracy of 64.06% and 57.42% for valence and arousal, respectively, as shown in Figure 5. The confusion matrix for the proposed method 3 illustrates the classification performance on the DEAP dataset, as shown in Figure 5a,b. The results suggest that the model performs well in correctly identifying low and high valence. The model has demonstrated a solid ability to accurately predict the valence class, with a true positive and negative rate of 64%. However, for the arousal class, the model exhibits a higher rate of false positives, with 48% of low-arousal trials being classified as high arousal.

**Table 1.** DEAP classification results.

| Class | Proposed Methods | | | KNN | CNN |
|---|---|---|---|---|---|
| | **M1** | **M2** | **M3** | **K = 5** | **EEGNet** |
| Valence | 58% | 63.5% | 64% | 59% | 57.24% |
| Arousal | 55.48% | 57.1% | 57.4% | 58% | 56.21% |

This table shows the classification results for the proposed methods compared to KNN and CNN (state-of-the-art machine learning algorithms for emotion recognition) on the DEAP dataset. The proposed methods are M1: MDRM, M2: MDRM plus PCA, M3: MDRM plus PCA plus hyperparameter tuning.

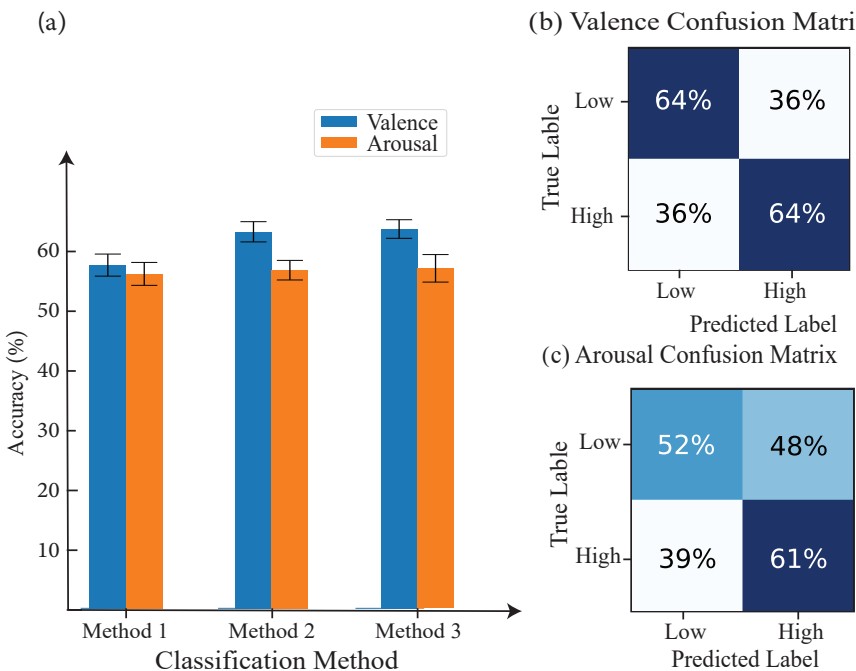

**Figure 5.** DEAP dataset classification results: (**a**) classification result comparison of the proposed methods. (**b**) Valence classification confusion matrix (method 3). (**c**) Arousal classification confusion matrix (method 3).

### 4.2. DREAMER Dataset

The performance comparison of the three proposed models on the DREAMER dataset is depicted in Figure 6. The confusion matrices for both valence and arousal classes are presented in Figure 6a,b. The results reveal a consistent pattern across both classes, with higher accuracy achieved in classifying high arousal and high valence, reaching 64% and 67%, respectively.

This finding suggests that the models may overemphasize certain features associated with high arousal and valence, leading to a potential bias in the classification results.

The accuracies achieved by the proposed methods are compared to those of CNN and KNN classification approaches in Table 2. The results indicate that the proposed method 3 achieves superior performance, achieving accuracies of 56% for valence and 58.64% for arousal, outperforming the state-of-the-art techniques.

**Table 2.** DREAMER classification results.

| Class | Proposed Methods | | | KNN | CNN |
|---|---|---|---|---|---|
| | M1 | M2 | M3 | K = 5 | EEGNet |
| Valence | 54.94% | 55.25% | 56% | 54.4% | 55.34% |
| Arousal | 52.16% | 54.32% | 58.64% | 58% | 56.27% |

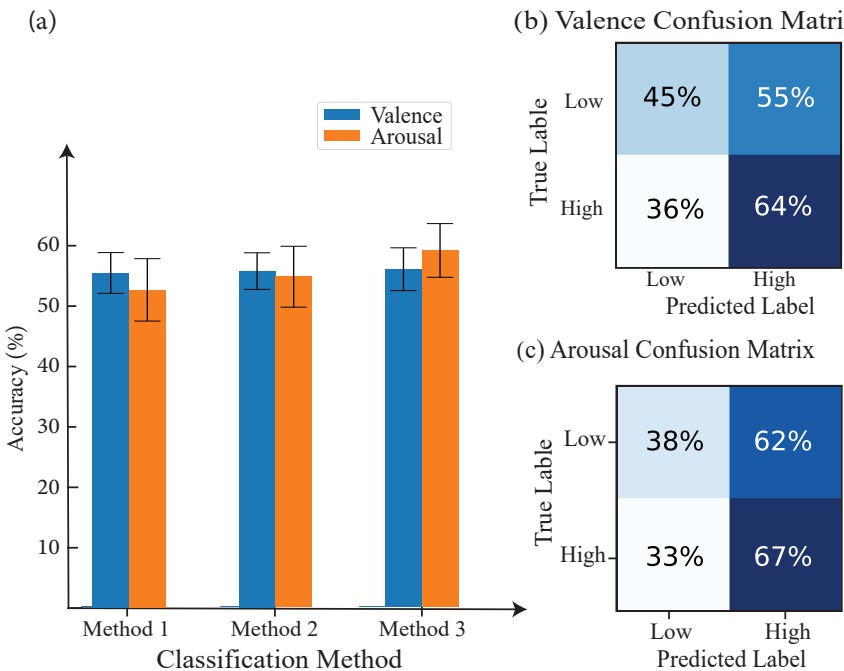

**Figure 6.** DREAMER Dataset classification results: (**a**) classification result comparison of the proposed methods. (**b**) Valence classification confusion matrix (method 3). (**c**) Arousal classification confusion matrix (method 3).

### 4.3. MAHNOB Dataset

The MAHNOB dataset was used to further evaluate the robustness of the proposed model. As shown in Figure 7, the proposed method exhibited a notable improvement in average accuracy using method 3.

A comparison of the proposed method's performance with KNN and CNN on the MAHNOB dataset is shown in Table 3. The results reveal that the proposed method 3, achieved an average accuracy of 56% and 60% for valence and arousal, respectively. These results are comparable to or higher than those of state-of-the-art methods.

**Table 3.** MAHNOB dataset classification results for proposed methods.

| Class | Proposed Methods | | | KNN | CNN |
|---|---|---|---|---|---|
| | M1 | M2 | M3 | K = 5 | EEGNet |
| Valence | 51% | 52.2% | 56% | 46% | 54.24% |
| Arousal | 56.4% | 57% | 60% | 41% | 60.62% |

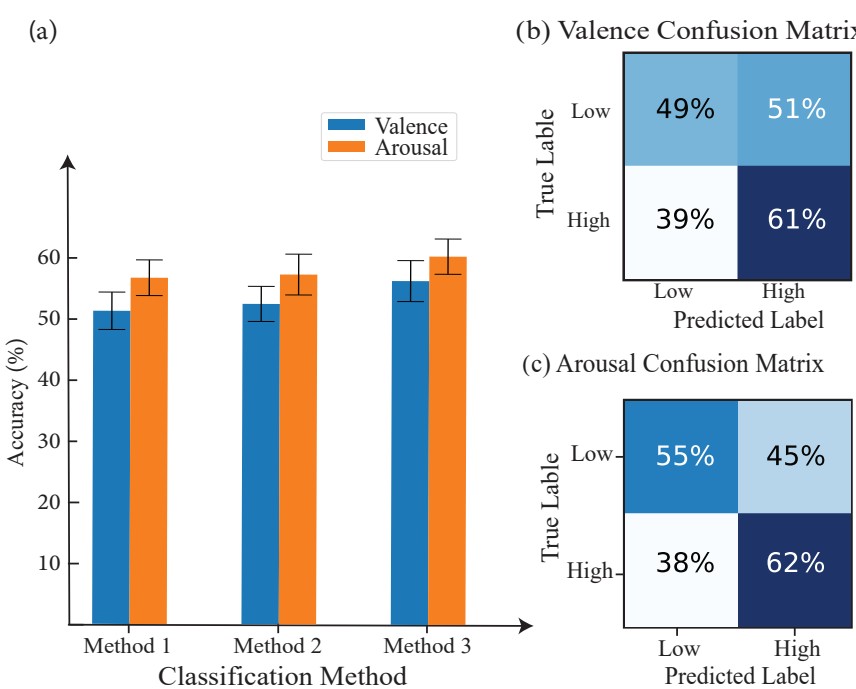

**Figure 7.** MAHNOB dataset classification results: (**a**) classification result comparison of the proposed methods. (**b**) Valence classification confusion matrix (method 3). (**c**) Arousal classification confusion matrix (method 3).

The accuracy of the classification results for high valence and arousal is demonstrated in the confusion matrix presented in Figure 7a,b, with an accuracy of 61% and 62%, respectively. The confusion matrix of the proposed method 3 exhibited a distinct diagonal pattern, indicating that the model accurately identified the participant's emotional state for the arousal class. However, for the valence class, the model exhibited higher precision for high valence. The precision for low valence was relatively lower, as the model misclassified some instances of low valence as high valence. These results suggest that the model is better at identifying high-valence emotions and may require further improvement to distinguish low-valence emotions more accurately.

### 4.4. SEED Dataset

The SEED dataset includes emotions categorized as positive, negative, and neutral across 15 trials. Table 4 displays the three-class classification results for the dataset. The results of comparing the proposed methods with KNN and CNN are illustrated in Table 4. The comparison shows that the proposed methods achieved comparable performance by utilizing the Riemannian framework in classification. The results show that the model's accuracy increased proportionally with the complexity of the model, resulting in a 12% improvement in accuracy compared to method 1, as indicated in Figure 8. The model's accuracy was also improved by integrating dynamic hyperparameter tuning and feature selection. This integration led to a drastic increase in accuracy.

**Table 4.** SEED classification results.

| Class | Proposed Methods | | | KNN | CNN |
|---|---|---|---|---|---|
| | **M1** | **M2** | **M3** | **K = 5** | **EEGNet** |
| Accuracy | 51% | 51.26% | 63.4% | 52% | 52.66% |

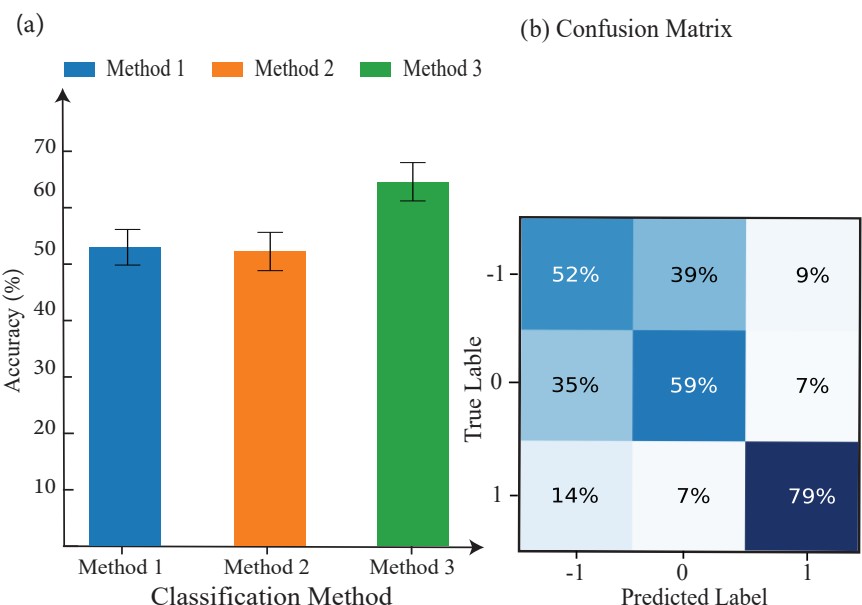

**Figure 8.** SEED dataset classification results. (**a**) Classification result comparison of the proposed methods. (**b**) Multi-class classification confusion matrix (method 3).

The classification of emotions in the SEED dataset presents several challenges because it involves mixed positive, neutral, and negative emotions, which require the model to distinguish between subtle emotional nuances. As shown in the confusion matrix in Figure 8b, the model produced more false positives (negative trials misclassified as neutral) than false negatives (neutral trials misclassified as negative). This finding suggests that the model may be overly sensitive to neutral stimuli and could benefit from a more nuanced understanding of the features that differentiate between neutral and negative emotions. Despite these challenges, the positive class achieved a satisfactory classification performance of 79% in terms of accuracy. This result indicates that the model is effective in recognizing positive emotions. Moreover, the improvements resulting from integrating dynamic hyperparameter tuning and feature selection demonstrate the overall effectiveness of the proposed approach for classifying emotions in the SEED dataset.

### 4.5. Overall Performance Comparison of Proposed Methods and Baseline Models on Multiple Datasets

The proposed methods' performance in terms of mean was compared with K-nearest neighbors (KNN) and the convolutional neural network (CNN) across DEAP, DREAMER, MAHNOB-HCI, and SEED datasets. The overall results, as shown in Figure 9, demonstrate that the proposed methods offer comparable or better performance compared to two commonly used classification methods. Among the three methods, i.e., M1 (MDRM only), M2 (MDRM plus PCA), and M3 (MDRM plus PCA plus hyperparameter tuning), M3 showed higher accuracy compared to M1 and M2, across all four datasets, as expected. Between M1 and M2, some datasets showed higher accuracy of M2 compared to M1 but overall the difference between them was moderate.

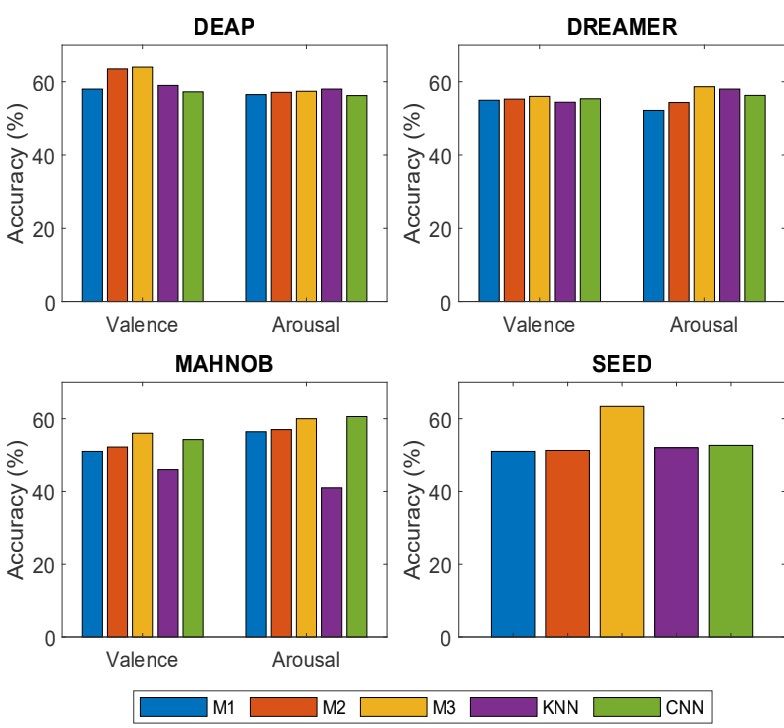

**Figure 9.** Performance (classification accuracy) comparison of the three methods proposed in this work, compared with the results from using KNN and CNN applied to the same datasets.

## 5. Discussion and Future Work

Understanding and achieving desirable results in the field of emotion recognition remains a challenging task, presenting opportunities for further innovation and investigation. Although previous studies [7,28,30,32,34,38] have demonstrated satisfactory results, there are still critical issues yet to be addressed, such as robustness, generalizability, and handling of intra-subject variability and observation variations. To address these gaps, this work aims to develop a robust and flexible pipeline for emotion recognition that can be applied to real-world data, while also mitigating the impact of these challenges. The proposed approach leverages Riemannian geometry to classify emotions by preprocessing raw EEG data and reducing its dimensionality using PCA [29]. The preprocessed data are represented as covariance matrices, on which Riemannian metrics and algorithms are applied for classification. The results showed that the utilization of Riemannian geometry alone improved the classification accuracy for the valence class compared to a CNN model on the DEAP dataset by 0.76%, and for the MAHNOB dataset, the accuracy improved by 6% compared to KNN. However, the accuracy for M1, the baseline model, remained slightly lower than that of state-of-the-art algorithms. To overcome this limitation, we introduced M2, which incorporates traditional PCA in addition to the MDRM approach. By integrating PCA, we aimed to capture relevant information and reduce noise in the EEG signal representations. The results across the DEAP, DREAMER, and MAHNOB datasets demonstrated improved accuracy compared to M1 as shown in Figure 9.

Building upon the improvements of M2, M3 introduced dynamic PCA, enabling adaptive feature extraction and a more effective representation of EEG signal variations. The results demonstrated that M3 consistently achieved higher accuracy compared to both M1 and M2. Notably, in the DEAP dataset, the accuracy increased by 6% and 2% for the valence and arousal classes, respectively. The DREAMER dataset exhibited a similar pattern, with accuracy improvements of 6.48% and 1.1% for the arousal and valence classes, respectively. In the MAHNOB dataset, M3 yielded accuracy improvements of 3.6% for arousal and 5% for valence. The SEED dataset showcased substantial enhancement, with a classification accuracy improvement of 12.4%. These consistent improvements across the datasets, combined with the notable performance compared to state-of-the-art machine learning

methods, particularly in the classification of the SEED dataset, underscore the effectiveness of our proposed approach. While M1 may yield lower accuracy compared to the state-of-the-art methods, it serves the purpose of establishing a baseline and showcasing the potential of Riemannian geometry in EEG signal analysis. The progression from M1 to M3 highlights the significance of integrating these methodologies to enhance classification performance.

To assess the generalizability of the model, we evaluated it across different datasets with diverse characteristics, including DEAP, DREAMER, SEED, and MAHNOB, demonstrating its robustness and adaptability for emotion classification. Furthermore, the results confirmed the effectiveness of using the Riemannian mean and AIRM in Riemannian space, as they are designed to operate on covariance matrices, making computations more efficient and accurate compared to other feature spaces [47]. These findings highlight the effectiveness of Riemannian geometry for emotion classification and pave the way for further brain–machine interface (BMI) research in this area.

While this study provides valuable insights, there are limitations to consider. The data used were pre-selected based on specific criteria to avoid imbalanced data, resulting in a potentially sparse dataset. Future research should focus on addressing these limitations by collecting more diverse and representative datasets and developing techniques to handle imbalanced or sparse data. Additionally, exploring dimensionality reduction directly on covariance matrices in Riemannian space and investigating different manifold-based classifiers would be worthwhile. Furthermore, integrating EEG signals with other physiological signals, such as electromyogram (EMG) and electrocardiogram (ECG), as well as non-physiological signals like facial expressions or body gestures, could further enhance the accuracy of emotion recognition. Therefore, our logical next step involves exploring these multimodal approaches and applying the method presented in this work.

## 6. Conclusions

This study contributes to the growing literature on EEG-based emotion classification, demonstrating the potential of Riemannian geometry as an effective approach. By investigating classification performance across diverse datasets, we address limitations in emotion recognition using Riemannian metrics and validate the robustness and real-world applicability of our proposed method. The key contributions are the incorporation of PCA-based dynamic feature extraction and the validation of the method's generalizability and robustness by outperforming state-of-the-art techniques on four EEG datasets with varying characteristics. Our findings suggest that hyperparameter tuning and dimensionality reduction techniques can further optimize model performance. Additionally, the effectiveness of using the Riemannian mean and AIRM in the Riemannian space for covariance matrix computations was supported, providing a more efficient and accurate approach. Successful implementation across different datasets with varying characteristics showcases the model's generalizability and adaptability. These results support future applications of Riemannian geometry in EEG classification research, paving the way for enhanced emotion recognition capabilities.

**Author Contributions:** Conceptualization, Z.A.-M. and J.-H.P.; methodology, Z.A.-M.; software, Z.A.-M.; validation, Z.A.-M. and N.B.; formal analysis, Z.A.-M.; investigation, Z.A.-M., R.C. and J.-H.P.; data curation, Z.A.-M. and I.F.K.; writing—original draft preparation, Z.A.-M.; writing—review and editing, Z.A.-M., N.B., I.F.K., R.C. and J.-H.P.; visualization, Z.A.-M.; supervision, J.-H.P.; project administration, J.-H.P. All authors have read and agreed to the published version of the manuscript.

**Funding:** This research received no external funding.

**Institutional Review Board Statement:** This study did not require ethics approval as it solely utilized publicly available datasets.

**Informed Consent Statement:** As this study utilized publicly available datasets, no specific consent to participate was required from individuals. Consent to publish the results of this study was obtained from the dataset providers in accordance with their terms of use and licensing agreements. All necessary permissions were acquired prior to the utilization of the datasets.

**Data Availability Statement:** The data used in this study, namely DEAP, DREAMER, SEED, and MAHNOB, are available upon request from the corresponding author and were obtained with proper permissions and approvals. Researchers interested in accessing these datasets can find them at the following sources: DEAP: https://www.eecs.qmul.ac.uk/mmv/datasets/deap/index.html (accessed on 1 October 2022), DREAMER: https://zenodo.org/record/546113 (accessed on 10 October 2022), SEED: https://bcmi.sjtu.edu.cn/home/seed/ (accessed on 2 November 2022), MAHNOB: https://mahnob-db.eu/hci-tagging/ (accessed on 20 September 2022).

**Conflicts of Interest:** The authors declare no conflict of interest.

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
