# Peer review of "The Efficacy and Utility of Lower-Dimensional Riemannian Geometry for EEG-Based Emotion Classification"

_applsci, doi:10.3390/app13148274_

Round 1

Reviewer 1 Report

This article Multi-modal multi-transformer for image fake news detection over social media proposed a introduces a method to improve accuracy in EEG-based emotion classification using the Riemannian manifold, applied to four well-known datasets (DEAP, DREAMER, SEED, and MAHNOB). Authors claim that their classification accuracy and robustness was compared with several state-of-the-art classification methods, which supports the validity and efficacy of using Riemannian geometry for enhancing the accuracy of

EEG-based emotion classification.

However, I have several suggestions that might improve the quality of the manuscript.

1.     It’s an interesting topic but unfortunately the organization and contributions Minimum Distance to Riemannian Mean (MDRM) is not explained properly.

2.     This paper provides a quite simple results based on different dataset. While a
thorough investigation between many data, or an improvement and combination
of these methodologies may provide some sort of scientific contribution, the
present study does not provide any strong novelty.

3.     Introduction should be written properly, and provide implications or motivation and organization of the work

Reviewer 2 Report

1. Justify the novelty of work. I aware that with M3, the accuracy is increased, but overall accuracy still remain low. Can the authors justify how significant to have such increment in accuracy. Taking the output for DEAP Dataset as an example, the increment of accuracy from M1 to M3 for both Valence and Arousal are low. How is the increment herein important in overall performance analysis?

2. Taking detection of Arousal (in DEAP, DREAMER, and MAHNOB dataset) & Accuracy (in SEED dataset) as an example, comparing accuracy obtained in M1 and CNN method, results in M1 are actually lower than that of the CNN method for DEAP, DREAMER, MAHNOB, and SEED dataset. Justify the role/ significant/ novelty of Minimum Distance to Riemannian Mean in this study. 

Reviewer 3 Report

Review To Applsci-2501328

This study was to demonstrate the efficacy and utility of Riemannian geometry applied to classification problems. This study uses three primary methods to investigate the impact of using solely Riemannian geometry for classification and the effect of employing dimensionality reduction and hyperparameter tuning on the classification results. This study suggests that incorporating hyper-parameter tuning and dimensionality reduction can further optimize the model’s performance. Dynamic feature selection and PCA-based dimensionality reduction were found to improve classification accuracy.

This study will be a good method to classify the EEG-based Emotion.

1. Indicate the study design in the research title.
2. Add study methods to the abstract. 

Round 2

Reviewer 1 Report

Can be accept in current form

Reviewer 2 Report

The author has addressed the concerns accordingly.